Title: Competing theories of probabilistic computations in the brain.

## Scientific question

How does neural activity represent probability distributions and how are probabilistic computations implemented in neural circuits?

## Background

Behavioral experiments consistently show that humans and animals reason probabilistically when presented with noisy, ambiguous, or incomplete information [Knill96]. This requires a neural representation of probability distributions and neural circuits capable of implementing the operations of probabilistic inference in a manner consistent with that representation. Currently, there exist three principal hypotheses:

1. The **probabilistic population code (PPC)** [Ma06, Beck08] assumes that different (possibly nonlinear) functions of neural activity encode the natural parameters of posterior distributions. Efficient use of a PPC predicts three ubiquitously observed cortical phenomena independent of the specific probabilistic computation (generative model) being performed: (1) fixed Fano factors, (2) amplitude encoding of confidence, and (3) linear evidence integration [Beck11]. From a computational perspective, PPCs have also been shown to be naturally incorporated into complex inference schemes based on Free Energy minimization [Friston10] as well as message passing and predictive coding schemes [Rao99, Beck12, Pitkow 17].

2. The **distributed distributional code (DDC)** [Zemel98,Sahani03,Pitkow12] also assumes that neural activity encodes uncertainty parametrically, like a PPC, but here the uncertainty is encoded as a set of expectations of non-linear encoding functions with respect to the represented distribution. These expectations can be thought of as generalised moments of the distribution or as the mean parameters of a corresponding exponential family distribution. Importantly, DDCs do not require that the represented distribution is from a tractable parametric family. The computational appeal of DDCs is that they are very well suited to such computations and translate the often intractable integrals to learning simple linear mappings. Recent work has shown that they can support accurate inference and learning in hierarchical generative models [Vertes18] and allow for generalising successor features and value function computation to partially observed settings [Vertes19]. Performing inference with DDCs in dynamical environments has been shown to account for psychophysics data related to perceptual phenomena called postdiction, i.e. inferences about past percepts [Wenliang19].

3. **Neural sampling codes** [Hoyer03, Fiser10] assume that neural activity represents samples from the represented posterior distributions. It directly corresponds to a

powerful and widely used class of machine learning algorithms (MCMC). Specific neural sampling models differ mainly in their choices for the mapping between random variables being represented and neural responses. Continuous latents could be represented by firing rates [Hoyer03], membrane potentials [Orban16, Echeveste19], or linear projections of instantaneous population firing rates [Savin14]; binary latents could be represented by spikes and post-synaptic potentials [Buesing11]. Sampling makes general predictions about the structure of neural variability and its link to uncertainty [Berkes10, Orban16] and was shown to explain neural response means and covariability in V1, as a function of stimulus features [Orban16] and behavioral task [Haefner16].

## Challenge or controversy

Even though these proposals have been around, in some form, for many years there is substantial debate and controversy regarding which is the best candidate for the cortical neural code. There are multiple reasons for this lack of consensus:

a. Differences in the inferred variables: while studies advocating for PPCs typically focus on inference over concrete task-relevant stimulus variables like orientation, neural sampling models typically focus on inference over abstract latents in a generative model of the inputs.

b. Differences in the language used to describe inferred variables, e.g. task-defined variables like orientation (PPC) compared to neural sampling and DDCs representing latent variables in a generative model of the stimulus (but distinct from it).

c. Lack of an agreed upon means to determine what variables are being represented in a given population of neurons.

d. There are no agreed upon experimental knobs to manipulate uncertainty about variables represented in a given area. Moreover, most studies often relied on assumptions about what manipulations change subjective uncertainty about task-defined variables.

e. Proponents of the different hypotheses typically compare their predictions against different aspects of neural data (e.g. means and variance across multiple brain areas during multisensory and evidence integration tasks for PPCs, means and covariances for neurons performing inference in image models with neural sampling).

f. Difference in the role of optimality principles: while PPCs stress statistical optimality from a probabilistic decoding perspective, neural sampling typically focuses on how to encode a posterior for use in a MCMC sampling-based inference algorithm.

g. The implicit, and possibly false, assumption that these hypotheses are universal in cortex, despite the fact that machine learning solutions suggest that different data structures and approximate inference algorithms may be better suited at different computational stages for optimal decision making.

h. The need for further assumptions beyond the nature of the representation to compare models to empirical data, and to each other. For instance, in special cases the very same system can be explained as performing sampling in a specific generative model while also interpreting the output through the lens of a PPC [Shivkumar18]. Similar relationships may exist between PPCs and DDCs, as well as DDCs and sampling,

suggesting that there might be equivalence classes that cut across the three main proposals considered here that are not yet understood.

Given these multiple formidable challenges it is hard for any one lab to address them on their own, instead requiring a larger collaborative effort as proposed here.

## Concrete outcomes

This central goal of this collaboration is to investigate the relationships between the currently proposed alternatives and, if possible, derive empirically testable differential predictions. Specific aims:

1. Create a common language by which to describe all proposals.
2. Lay out the key intuitions and assumptions behind each framework.
3. Derive commonalities and differences leading to either a Venn diagram of proposals, or, if appropriate, a space of models in which the different proposals occupy different parts.
4. Agree on a list of desiderata for a neural code for distributions (e.g. flexibility, learnability, efficiency, time/neuron constraints).
5. If possible, identify predictions for each coding framework that are independent of the generative model and the details of the inference algorithm employed.
6. Identify potential implications for plasticity and learning in neural circuits.
7. Design critical experiments to differentiate between mutually exclusive aspects of the proposals.

## Benefits to the community

1. A deeper understanding of the existing proposals for how probabilities are represented by neural responses will allow us to design experiments to falsify any or all of them.
2. A major advance in our understanding of the nature of the neural code and the applicability of Bayesian inference framework for understanding neural computation will provide potential inspiration for representations and inference algorithms used in AI.

## Roles of core members

**Jeff Beck (PPC)**

Jeff was one of the original developers of theoretical justifications for PPCs and has investigated them in a wide variety of computational contexts such as cue combination, evidence integration, visual search, coordinate transformation, object tracking, odor demixing, and blind source separation [Ma06, Beck08, Beck11, Beck12].

**Ralf Haefner (neural sampling)**

Ralf has developed neural sampling based models to explain neural and psychophysical data and compared them to alternative representation schemes like PPCs and DDCs [Haefner16, Shivkumar18, Lange20a, Lange20b].

**Xaq Pitkow (PPC/DDC)**

Xaq has developed generalizations of PPCs and DDCs to multivariate probabilistic graphical models [Pitkow12, Raju16], and analysis frameworks that can identify the nonlinear computational dynamics implied by neural data [Pitkow17]. He also connects these perceptual models to rational action planning through closed-loop control [Raju16, Pitkow17, Wu20].

**Cristina Savin (neural sampling)**

Cristina works on probabilistic computation in neural circuits, with a focus on learnable probabilistic representations [Munk18], and probabilistic inference in memory systems [Savin14b]. She has developed a spike-based distributed sampling scheme [Savin14] and new hypothesis testing methods for generative-model-agnostic sampling predictions [Fiser13].

**Eszter Vértes (DDC)**

Eszter has worked on combining DDCs with algorithms for learning generative models and learning to perform inference in them [Vertes18]. She has further shown that DDCs allow for generalising successor features to partially observable settings [Vertes19].

## Senior advisors:

**Mate Lengyel (sampling)**
**Alex Pouget (PPC)**
**Rajesh Rao (predictive coding)**
**Maneesh Sahani (DDC)**

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

**All core members agree to the requirements outlined in the call for proposals:**

- Incorporating feedback from the community and potentially welcoming new CCN community members to the GAC based on their written commentary to the GAC proposal
- Running an online kickoff workshop for CCN2020, inclusive of both founding core GAC members and those new members who joined through the community feedback process
- Writing the position paper to be submitted ~December 2020 to a curated special issue, to be accompanied by commentary pieces authored by attendees of the CCN2020 kickoff workshop
- Attending and presenting progress at the following CCN2021

In place of signatures:

*Ralf Haefner*
*Jeff Beck*
*Xaq Pitkow*
*Cristina Savin*
*Eszter Vertes*