# OpenReview forum: "Competing theories of probabilistic computations in the brain."
_ccneuro.org/CCN/2020/Workshop/GAC_

### Official Review · ~Dylan_Festa1 · 2020-08-24
**Excellent proposal, entirely in the spirit and scope of CCN GAC**

**Rating:** 10
**Soundness:** Strongly agree
**Confidence:** 5

**Review:**

## Introduction

Psychophysical and psychological experiments often seem to indicate that humans, and several species of non-human animals, can take decisions or regulate actions following probabilistic forms of reasoning. In other words, they perform operations of probabilistic inference that account for sensory ambiguity or prior knowledge.

The authors propose a collaborative investigation on the neural mechanisms that make these operations possible. They identify three possible candidate principles, and compile a thorough list of challenges and controversies surrounding the three.

Finally, they list the concrete outcomes of such collaboration.

##  Quality

I consider this proposal well formulated, precise, interesting and thorough. I particularly appreciate the long list of challenges, which reveals both the authors' expertise, and their awareness of the difficulties and nuances that surround the issue.

In my opinion the quality is outstanding. Diverse teams of scientists, who have been working at the core of these issues for several years, decided to collaborate together to build a common language and identify common solutions.

## Clarity

The text is extremely clear and well written. References are precise and always on point.

## Originality

The question of how the brain may produce probabilistic computations has been discussed for several years. I would say that the originality resides in the idea of collaborating for a common language and shared definitions, with great awareness of all the possible hindrances and limitations.

## Significance

The question examined here is quite central for neural coding. As stated in the "Benefits" section, knowing what neurons are representing, and how, may lead to better representations and algorithms for AI. I would add that better decoding would improve performance of neuroprosthetics, such as cochlear implants, or brain-machines interfaces.

### Pros

+ Relevant for the community
+ Authors have unquestionable expertise
+ May inform and drive future research, avoiding the risk of misconceptions, campanilism, and groups talking past each other

### Cons

The only issue I see is that, as mentioned in the text, the challenges are "formidable" and not necessarily easy to overcome, with our current theoretical, computational and experimental tools.

Many of the "concrete outcomes" listed appear either very preliminary, or somehow general. However I think that this reflects the author's intellectual honesty, and the need for such work.



**Comments:**

This review process somehow confuses me: with only 6 proposals in total, is there a need for reviews from the broader community? (besides the public comments and the organizer's panel?)

(ok, I just realized that reviews also seem to be posted as public comments... which answers at least in part the previous question)


**Controversy:**

Strongly agree

**Definition:**

Agree

**Expertise:**

Strongly agree

**Outcomes:**

Strongly agree

---

> ### Public Comment · ~Ralf_Haefner1 · 2020-09-07
> **Brief response**
>
> Thank you for the encouraging feedback.
>
> “Many of the "concrete outcomes" listed appear either very preliminary, or somehow general. However I think that this reflects the author's intellectual honesty, and the need for such work.”
>
> We agree and point to our specific aim #7 which is to “Design critical experiments to differentiate between mutually exclusive aspects of the proposals.” Being successful with this most ambitious outcome requires substantial progress on the more preliminary steps laid out (aims #1-6).

---

### Official Review · ~Thomas_Schatz1 · 2020-08-26
**Bringing theories of probabilistic computations in the brain closer to the test**

**Rating:** 9
**Soundness:** Strongly agree
**Confidence:** 4

**Review:**

In my opinion this is an excellent proposal for a GAC.

The question addressed is that of comparing, integrating, contrasting and ultimately testing existing theories of probabilistic neural computations that have mostly been considered separately so far. This is a question at the interface of cognition, computations and neuroscience and of obvious interest to the CCN community.

The proposal clearly identifies three competing viewpoints (probabilistic population codes, distributed distribution codes and neural sampling codes) in the literature and the team features recognized experts on each of those viewpoints. Addressing the proposed question is likely to require a tight collaboration between researchers in different labs that have been mostly working on only one or two of the existing theories, making it an excellent GAC proposal.

The main risk I can see for this project is a dispersal of efforts, as the question remains quite open-ended at this point. I think working out a concrete, detailed workplan is going to be challenging but crucial for the success of this proposal, and will require a tight and focused collaboration between the team members.



**Comments:**

Strengths
- well-defined question
- of clear relevance to the CCN community
- clearly identified competing viewpoints
- team featuring recognized experts on each viewpoint
- significant progress appears possible in the proposed timeframe

Weaknesses
- the question is quite open-ended and there is a significant risk of dispersal in efforts


**Controversy:**

Strongly agree

**Definition:**

Strongly agree

**Expertise:**

Strongly agree

**Outcomes:**

Strongly agree

---

> ### Public Comment · ~Ralf_Haefner1 · 2020-09-07
> **Brief response**
>
> Thank you for the encouraging feedback.
>
> “The main risk I can see for this project is a dispersal of efforts, as the question remains quite open-ended at this point. I think working out a concrete, detailed work plan is going to be challenging but crucial for the success of this proposal, and will require a tight and focused collaboration between the team members.”
>
> We completely agree. One of the main purposes of the workshop would be to identify the most productive context within which to focus our efforts. This might be by all participating groups focusing on the same empirical dataset on the basis of which the competing model classes are compared. Such a dataset might consist of neural responses from a single brain area and single task, or multiple brain areas and tasks if necessary.

---

### Public Comment · ~Garrison_W._Cottrell1 · 2020-08-22
**Great question**

This is an excellent question, and it seems like one that actually could be answered. The goal, which I take to be to create a common, agreed-upon vocabulary and measures of success, so that these three approaches could actually be made "commensurate", so that the different models could be placed in competition and perhaps falsified, would be a great contribution to the field.

---

### Public Comment · ~Emin_Orhan1 · 2020-09-01
**question too general to be meaningful**

In my opinion, the question motivating this proposal ("How does neural activity represent probability distributions and how are probabilistic computations implemented in neural circuits?") is too general to be meaningful: probability distributions over what, probabilistic computations of what? There's no reason to think that the brain (or neural networks, in general) uses the same type of computation to marginalize out the brightness of a visual scene, let's say, as it does to marginalize out background objects to recognize a foreground object. Yes, marginalization is a well-defined concept in statistical theory, but there's no good reason to think that well-defined concepts in our cherished theories should correspond one-to-one to computations in the brain (or in neural networks, in general).

Given some noisy responses r in the periphery, neural networks will preserve the information in r under pretty general conditions (similarly for sampling, under pretty general conditions, noisy neural responses can always be interpreted as approximate samples from a posterior distribution, e.g. via SGLD). The task will dictate which moments of which latent variables need to be represented more explicitly vs. implicitly (most of these "latents" will not be interpretable for natural tasks). The precise details of which moments of which latents are approximated how will again depend on the specifics of the task and the model architecture. So, there's really no mystery about any of these things at a general level. So, I struggle a bit to understand what more the authors are hoping to learn at this level of generality.

---

> ### Public Comment · ~Ralf_Haefner1 · 2020-09-07
> **Brief response**
>
> The commentator raises several important questions, most of which are addressed in our proposal as challenges that have so far hindered a conclusive model comparison of existing theories of probabilistic computations in the brain. The key question added by the commentator concerns the general premise: Is it even reasonable to expect that the mathematical language of probabilistic inference can contribute materially to our understanding of the brain? While we disagree with the commentator and elaborate on our reasons below, this is ultimately an empirical question: Are there non-probabilistic theories and models that provide a more parsimonious explanation for empirical observations? In the spirit of Strong Inference [1] and model comparison underlying GACs we would invite critics of the Bayesian approach to understanding neural activity to present alternative proposals in our GAC workshop.
>
> The commentator raised another point, namely that it might not be possible to answer a question about representation, or computation, in isolation, but that it is important to compare pairs (is probabilistic model X1 with representation Y1 and algorithm Z1 a better explanation of the empirical data than probabilistic model X2 with representation Y2 and algorithm Z2?). We completely agree with this point and it is explicitly contained in reason (h) in our proposal. In fact this was the motivation behind our title, which emphasizes probabilistic computations rather than merely representations.
>
> Why do we believe the language of probabilistic inference to be a promising one to explain neural responses?
>
> The commentator raises reasonable and important questions about whether we should expect any unifying theory of probabilistic brain computation at all. Old concerns state that the brain is no more than a bag of tricks, so it is futile to search for unifying principles [2]. As a case study, attempts to provide interpretability to deep networks in machine learning struggle. It is unclear why the brain would be any more interpretable or lawful than a fully observable machine. The concern is that maybe, even if probabilistic inference does describe an animal’s behavior, there is no mechanism that is shared across neurons or brain areas or tasks.
> However, amidst its complexity, brain circuitry has repeating motifs and heritable architecture and learning rules that imbue neural computation with a strong but unknown inductive bias. The no-free-lunch theorem [3] shows that no architecture can be universally better for all possible worlds. Since brains evolved in one particular world, it stands to reason that there may be a relationship between the common structure of natural tasks and the common neural motifs that favor good solutions of them.
> Since the world ultimately rewards behavior more than internal brain representations, it is not immediately apparent why the brain might create lawful general purpose representations, as opposed to simply mapping observations idiosyncratically to actions. However, given the natural separation between the world variables to be controlled and the means of controlling them, modularity can arise as a general principle. Furthermore, actions must be taken in a world of hidden variables, and probabilistic inference is the best way to identify these hidden variables, so we argue that it is reasonable that a bias favoring some generalizable form of probabilistic computations would bestow added fitness upon a brain. This might or might not be simply recognizable within individual neurons; a unifying description may only emerge at a higher level of abstraction, say, at the level of neural manifolds.
> This commentator is himself an author of a paper that identified how probabilistic computation emerges automatically in multiple, naturally modular tasks [4]. As he correctly states, “The precise details of which moments of which latents are approximated how will again depend on the specifics of the task and the model architecture.” Indeed we have acquired, and continue to acquire, empirical knowledge of these specifics. Yet even without this knowledge, we can fruitfully study which representations emerge in which architectures and which tasks to generate testable predictions for neuroscience and machine learning.
> Our goal here is not to conclusively answer these questions but rather to refine the framework within which these questions can be answered, by making the assumptions more explicit, the connections between theories more apparent, and their predictions more discriminating. When those assumptions are unpacked, we may find more similarities than commonly understood [5].

---

> > ### Public Comment · ~Ralf_Haefner1 · 2020-09-07
> > **Brief response (references)**
> >
> > [1] John R. Platt (1964). Strong inference. Science. 146 (3642): 347–53. doi:10.1126/science.146.3642.347.
> > [2] V.S. Ramachandran (1985). The neurobiology of perception. pp 97–103.
> > [3] David H. Wolpert (1996). The Lack of A Priori Distinctions Between Learning Algorithms. Neural Computation 1996 8:7, 1341-1390
> > [4] Emin Orhan, A., & Ma, W. J. (2017). Efficient probabilistic inference in generic neural networks trained with non-probabilistic feedback. Nature Communications, 8(1). https://doi.org/10.1038/s41467-017-00181-8
> > [5] Shivkumar S, Lange L, Chattoraj A, Haefner R (2018). A probabilistic population code based on neural samples. Advances in Neural Information Processing Systems.